# Numerical-Experimental Plastic-Damage Characterisation of Additively Manufactured 18Ni300 Maraging Steel by Means of Multiaxial Double-Notched Specimens

**Tiago Silva** [1], **Afonso Gregório** [2], **Filipe Silva** [1], **José Xavier** [3,*], **Ana Reis** [1,4], **Pedro Rosa** [2] **and Abílio de Jesus** [1,4]

1 INEGI, Instituto de Ciência e Inovação em Engenharia Mecânica e Engenharia Industrial, Rua Dr. Roberto Frias, 4200-465 Porto, Portugal; tesilva@inegi.up.pt (T.S.); fsilva@inegi.up.pt (F.S.); arlr@fe.up.pt (A.R.); ajesus@fe.up.pt (A.d.J.)
2 IDMEC, Instituto Superior Técnico, Universidade de Lisboa, Av. Rovisco Pais 1, 1049-001 Lisboa, Portugal; afonsogregorio@tecnico.ulisboa.pt (A.G.); pedro.rosa@ist.utl.pt (P.R.)
3 UNIDEMI, Department of Mechanical and Industrial Engineering, NOVA School of Science and Technology, NOVA University Lisbon, 2829-516 Caparica, Portugal
4 FEUP, Faculdade de Engenharia da Universidade do Porto, Rua Dr. Roberto Frias, 4200-465 Porto, Portugal
* Correspondence: jmc.xavier@fct.unl.pt

**Abstract:** Additive manufacturing (AM) has become a viable option for producing structural parts with a high degree of geometrical complexity. Despite such trend, accurate material properties, under diversified testing conditions, are scarce or practically non-existent for the most recent additively manufactured (AMed) materials. Such data gap may compromise component performance design, through numerical simulation, especially enhanced by topological optimisation of AMed components. This study aimed at a comprehensive characterisation of laser powder bed fusion as-built 18Ni300 maraging steel and its systematic comparison to the conventional counterpart. Multiaxial double-notched specimens demonstrated a successful depiction of both plastic and damage behaviour under different stress states. Tensile specimens with distinct notch configurations were also used for high stress triaxiality range characterisation. This study demonstrates that the multiaxial double-notched specimens constitute a viable option towards the inverse plastic behaviour calibration of high-strength additively manufactured steels in distinct state of stress conditions. AMed maraging steel exhibited higher strength and lower ductility than the conventional material.

**Keywords:** 18Ni300 maraging steel; additive manufacturing; materials characterisation; numerical simulation; digital image correlation

## 1. Introduction

Over the last decade, additive manufacturing (AM) has been expanded towards the production of functional parts, aiming at outstanding performance compared with their conventional counterparts for structural applications [1,2]. The less restrictive limitations in geometrical design of parts (relative to other manufacturing processes) along with topological optimisation techniques has attracted the most varied industrial fields. Kumar and Nair [3] highlighted the usage of topologically optimised additively manufactured (AMed) parts as well as the replacement of twenty-part assemblies by a single geometrically complex component, in the aerospace industry. Meng et al. [4] indicated some efforts in using AMed parts as reinforcement in automotive structural assemblies. In medical applications, metallic porous parts are used as bone scaffolds [5], and dental prostheses are efficiently produced [6] with a high degree of customisation. Mazur et al. [7] illustrated the enhanced cooling rates of mould inserts with conformal channel systems in tooling, indicating decreased cycle times and improved life of the tooling system.

The reliability of high strength steels is based on the principle of a good compromise between strength and ductility. Owing to its convenient good weldability (enhancing printability) and excellent strength–ductility ratio, the maraging steels, namely the grade 18Ni300 alloy, have been gaining popularity within the relatively limited range of available AMed engineering materials, especially for the laser powder bed fusion (LPBF) technology. However, the promising outlook of LPBF is notorious [8] and well illustrated by the considerable amount of distinct manufacturers' solutions as well as the wider choice of materials (relative to other AM processes) including data regarding their processing. Sames et al. [9] and Lewandowski and Seifi [10] reported that the tensile and yield strengths of AMed steel alloys are identical to those of conventional manufacture (CM). In fact, in particular for the LPBF 18Ni300 maraging steel, the yield and tensile strengths can even outperform those of CM [11]. Even though that outstanding mechanical strength usually comes at the cost of lower ductility, in some cases, the maraging steel can be both stronger and more ductile [12,13] than the conventional counterpart, making it a very suitable candidate towards high-performance structural applications.

Despite the considerable development of industrial AM solutions, the experimental research has been typically focused on mechanical uniaxial tests. The limited data pave the way for a more comprehensive knowledge on the behaviour of AMed metallic alloys, through multiaxial, impact, fatigue and fracture toughness testing, using the most recent methods employed in conventionally manufactured (CMed) materials [14–16]. An enhanced characterisation is especially relevant in additive manufacturing due to the novelty of the processed materials, as well as for the maraging steel in particular, due to the reported strength sensitivity to loading condition [17,18]. Understanding the materials stress–strain (plastic) and damage behaviours under the most varied loading conditions is essential for the development of material databases that allow, for instance, numerical simulation validation of structural applications. Distinct specimen approaches can be used towards the calibration of state-of-stress sensitive models. This can be performed in a very thorough, yet long-winded manner, through the usage of a multitude of specimens, such as distinctly notched tensile and compression specimens, or, alternatively, through the usage a single specimen geometry (butterfly specimens) that is submitted to different (biaxial) loading conditions [19,20] as for example provided by an Arcan gripping system [21–23]. The intricate loading devices associated to the previous alternatives have motivated the usage of double-notched specimens with slightly different notch configurations [24], allowing for multiaxial testing in the (relatively) simple tensile and compression testing setups.

This study focused on the determination of plasticity-damage constitutive behaviour capable of accurately describing the mechanical response of the 18Ni300 maraging steel in distinct state of stress conditions. These data have been scarcely explored in AMed high strength steels. The influence of the metallurgical condition on the multiaxial mechanical strength of the alloys was also assessed. Double-notched (multiaxial) specimens of both AMed and CMed maraging steels were tested. Appropriate constitutive plastic-damage models were identified by inverse experimental-numerical calibration of the tested specimens. Digital image correlation (DIC) was applied in order to enable the strain fields reconstruction over a whole region of interest, leveraging the comparison over the mechanical response between the numerical simulations and experimental data, and therefore strengthening the inverse identification process adopted in this research. In addition, DIC enabled to infer on damage initiation and, thus, the definition of a damage initiation model. For completeness of the study, tensile tests were performed for notched and smooth specimen configurations which enabled the damage initiation at high triaxialities range. Moreover, the tensile stress–strain curves dictated the initial guess (for incipient strain) of the inversely identified flow stress.

## 2. Materials and Methods

### 2.1. Additively Manufactured 18Ni300

Both AMed and CMed maraging steels were selected for this research. Whereas the conventional material was cast and subsequently vacuum remelted [25], the latter was processed though laser powder bed fusion (LPBF). A Yb fiber laser with a spot size of 70 µm was employed, using optimised process parameters towards ideal printing conditions as well as density and mechanical strength maximisation: laser power, $P_L$, of 400 W; scan speed, $v_c$, of 0.86 m/s; hatch spacing, $h_s$, of 95 µm; layer thickness, $d_s$, of 40 µm; particle size distribution ($d_{10}/d_{90}$) of 15/45 µm. In order to verify the materials compliance with the standard, their chemical compositions were analysed using spark emission spectroscopy. Table 1 displays the measured data for both metallurgical conditions as well as the relevant standard thresholds. Despite the slight differences in chemical composition, both metallurgical conditions are in accordance with the standard. Rather than the typical high carbon content, the martensitic microstructure of 18Ni300 steel is achieved by nickel addition, which is the main alloying element. Corrosion resistance due to carbide precipitation can be significantly compromised with local heating and subsequent cooling effect [26] (such as in welding or AM). Lowering the carbon content of the alloy can minimise such effect, highlighting the suitability of maraging steels towards AM processing [27,28]. It is worth noticing the significantly lower weight percentage of the Mn alloying element in the AM LPBF. This is potentially related with the high vapour pressure of such element, which may volatilise during powder (re)fusion processes.

**Table 1.** Chemical composition (% wt.) of the 18Ni300 steel in both metallurgical conditions.

|      | Ni        | Co      | Mo      | Ti      | Si     | Mn     | C      | P       | S       |
| ---- | --------- | ------- | ------- | ------- | ------ | ------ | ------ | ------- | ------- |
| [29] | 18.0–19.0 | 8.5–9.5 | 4.6–5.2 | 0.5–0.8 | <0.10  | <0.10  | <0.03  | <0.01   | <0.01   |
| AM   | 18.80     | 8.84    | 5.15    | 0.65    | 0.05   | 0.03   | 0.02   | <0.001  | <0.001  |
| CM   | 18.93     | 8.92    | 4.88    | 0.77    | 0.02   | 0.03   | 0.01   | <0.001  | <0.001  |

The micrograph cross-section method for porosity measurement was adopted in this study. AMed samples were cut in two distinct directions (perpendicular and parallel to the build directions) and embedded in resin (through compression mount). Manual and semi-automatic grinding using sandpaper, followed by polishing of the cut cross-sections (Struers Pedemax-2), allowed for its optical microscope observation (Olympus PMG3 and Zeiss Axiophot). Etchant is typically not used in these samples, given that the goal is to simply reveal the absence of material (pores) in a polished surface. Porosity was measured through defect image analysis. Through the comparison of the most popular porosity measurement techniques in AMed alloys [30], it was concluded that the differences between micrograph cross-section technique and the Archimedes method were approximately 1% for relative densities higher than 98% and using 4 distinct micrograph samples with 40% magnification. Moreover, this method has the advantage of revealing porosity shape and its distribution in the cross-section, which may enable the identification of process-related flaws (for instance, porosity alignment due to excessive scan speed). The microstructure analysis of the material samples was carried using the same previously described optical equipment, for both metallurgical conditions of the considered material. Samples were distinctly etched according to the observation purpose. Whereas grain morphology can easily be observed with chemical etching (nital 2%), melt pool geometry and AM-related microscopic features, such as laser trace, require electrolytical etching (oxalic acid at 6V DC for 50 s).

## 2.2. Compression of Multiaxial Double-Notched Specimens

For the identification of the pressure-dependent plastic and damage behaviours, the plane strain specimen specially designed in [24] was employed. The major advantages of this test method is its ability to explore an intermediate range of stress triaxialities, which can be rather challenging to achieve by using typical tensile or compression specimens. The specimen configuration consisted of a symmetric double-notched geometry, with a nominal ligament of 2 mm, as shown in Figure 1. It is important to note that the AMed specimens were machined to the final geometry in order to ensure tight geometrical and dimensional tolerances as well as good surface finishing. By selecting the pressure angle of the specimen (defined by the centre of the two holes orientation), the notch configuration is altered, allowing for different stress triaxialities. Even though the specimens can be loaded in both tensile or compressive loading conditions, the latter was preferred due to setup simplification. The tests were carried at an Instron 5900R testing machine making use of a 100 kN load cell. A quasi-static compression speed of 1 mm/min was imposed on the machine cross-head and two repetitions were conducted in order to ensure experimental repeatability. Due to specimen symmetry, it was important to ensure not only that the compression platens were oriented parallel to each other (top and bottom), but also perpendicular to the crosshead displacement direction. Regardless of load condition, for a pressure angle of 90° (refer to Figure 1b), no lateral stress is developed, meaning minimum stress triaxiality and theoretically pure shear conditions. Increasing the pressure angle (refer to Figure 1a) results in a combination of a tensile and shear state of stress, yielding intermediate positive stress triaxialities. By decreasing the pressure angle (see Figure 1c), a mixture of compression and shear is attained, developing intermediate negative stress triaxialities.

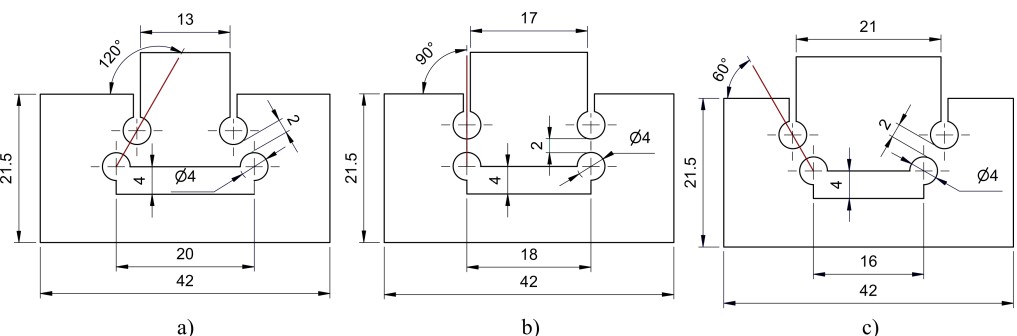

**Figure 1.** Multiaxial double-notched specimens for distinct pressure angle configurations: (**a**) 120°; (**b**) 90°; (**c**) 60° (units: mm).

An inverse experimental-numerical identification approach was employed in order to determine the plastic and damage constitutive parameters governing the mechanical response of the considered material [31]. In order to ensure the correct representation of the build numerical models, DIC was used (see Figure 2) [32,33]. It enables not only a comparison of the strain fields but also a more clear insight of the damage onset location, which, due to the material's expected fracture sensitivity to stress triaxiality, might depend on the specimen's pressure angle. A speckle pattern was applied using a IWATA Custom Micron CM-B airbrush (and an air compressor) through the dispersion of black paint on the previously homogenised and contrast-corrected specimen surfaces (using matte white paint) [34]. The pattern was applied on approximately half of the specimens, as shown in Figure 2d, since only one of the notches was monitored through DIC in order to enhance spatial resolution. In this study, a charged-coupled device (CCD) Manta G-1236 digital camera (8 bits, 4112 × 3008 pixels) coupled to a telecentric lens and a lighting device was used, as illustrated in Figure 2b. Correlated Solutions Vic-2D software was used for image correlation and reconstruction of the strain fields. The parameters were carefully selected based on a convergence parametric study, since in this application high strain gradients are expected in a very narrow region of the ligament [35]. Subset size was calculated to ensure a optimal match confidence of 0.01 pixel for a given assumed noise level. In particular, a

subset size of 97 pixels and a step size of 10 pixels were selected in a compromise between spatial resolution and resolution. Besides, a strain window of 15 data points with an algorithmic tensor definitions was selected.

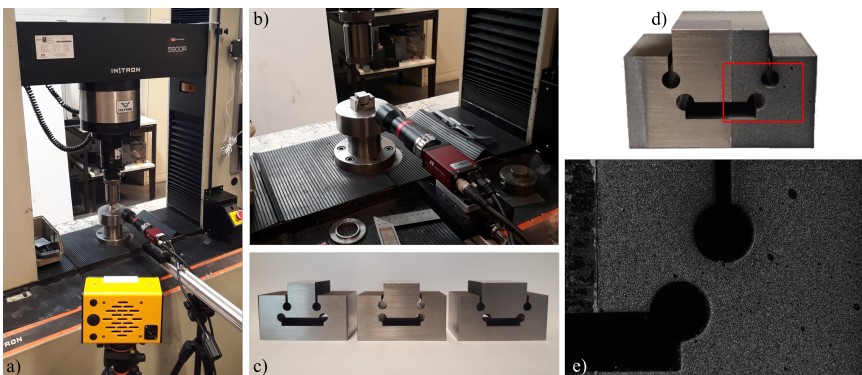

**Figure 2.** (**a**) The experimental apparatus of the multiaxial double-notched tests; (**b**) a detailed perspective of the DIC setup; (**c**) a sample of each tested specimen geometry; (**d**) speckle application on half-specimen; (**e**) DIC image of the signalled notch region.

Abaqus/CAE software with implicit analysis was used, considering an elastoplastic approach with von Mises (J2) yield criterion and isotropic hardening. The numerical simulation of the double-notched specimens was performed using a 2D plane strain approach. Their symmetry enabled the modelling of half-specimen. Four-noded elements with reduced integration (CPE4R) and an element size mesh of 0.05 mm were used. The model, as illustrated in Figure 3, consists of a top die (with one degree of freedom in y direction) that compresses the specimen onto a bottom die (encastred). Both dies were modelled as rigid bodies and interface friction was set to 0.2. Figure 3b,c shows the strain and stress localisation at the notch, for an incipient stage of the numerical simulation. Details about the used constitutive model are presented below.

### 2.3. Smooth and Notched Tensile Specimens

Tensile tests were conducted in order to depict the fracture behaviour of the considered materials under positive stress triaxialities. Three cylindrical geometries were tested: two notched and one smooth (see Figure 4b and Table 2). The analytical predictions proposed by Bridgman [36] were used in order to estimate initial stress triaxialities as a function of notch geometry. Threaded grips enabled holding both specimen ends to the testing machine. The tests were performed under quasi-static conditions (1 mm/s of pulling speed) in a servo-hydraulic testing machine (MTS 100 kN). An extensometer with 25 mm gauge length was used (MTS 632.12C-20) in order to measure local displacements (refer to Figure 4c). Load and displacement were measured for each sample, which were then plotted in terms of true stress and true strain.

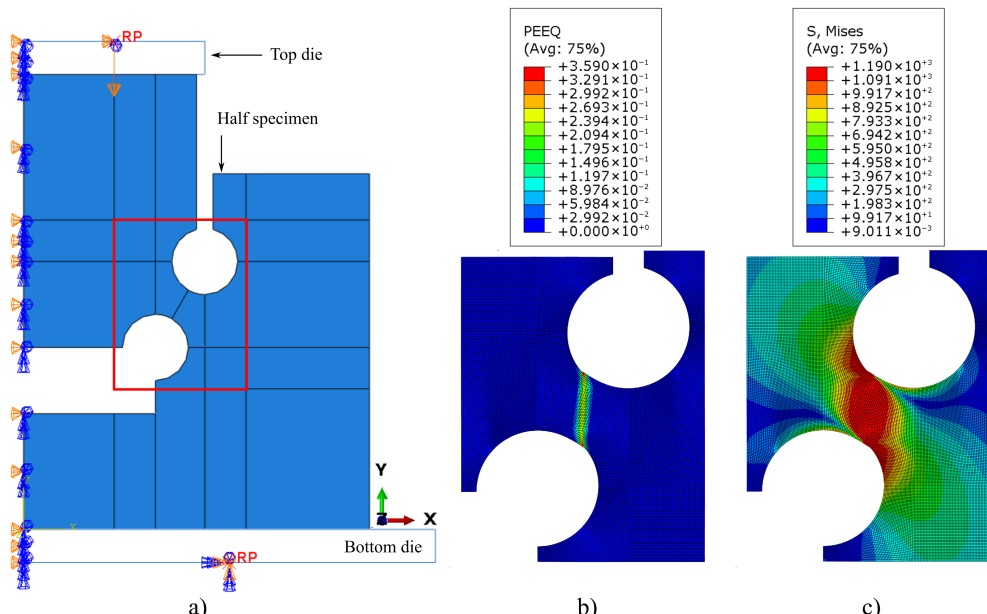

**Figure 3.** (**a**) Multiaxial specimen numerical model scheme with applied boundary conditions; (**b**) equivalent plastic strain; (**c**) von Mises stress fields of the red-signalled region for a top die displacement of 0.22 mm.

The usage of numerical modelling towards the identification of fracture initiation is a widely used methodology to circumvent plastic instabilities that hinder their direct estimation [37]. This enabled the identification of a damage behaviour as a function of the distinct pressure conditions, forced by the applied notches.

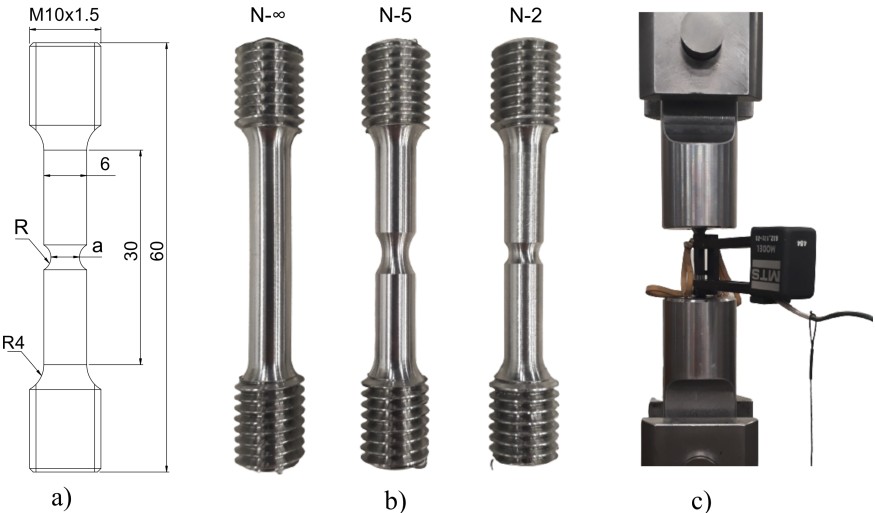

**Figure 4.** (**a**) The tensile specimen parameterised geometry; (**b**) example of the tested tensile samples; (**c**) the testing setup on servo-hydraulic machine with placed extensometer on specimen.

Four-noded axisymmetric elements with reduced integration (CAX4R) were applied to mesh the specimens with an element size of 0.1 mm. Figure 5a illustrates the adopted boundary conditions which include (apart from axisymmetry) zero $y$ displacement in the bottom nodes and a vertical displacement on the top nodes. The numerical models were built for the $L_0$ extensometer length, which was 25 mm. Figure 5b,c illustrates the typical field distributions of equivalent plastic strain and stress triaxiality of a notched specimen (AM N-5). It is observed that both are localised in the centre region of the specimen, where they are maximum. That location was therefore selected for data retrieval as regards the failure strains and respective stress triaxialities.

**Table 2.** Tested tensile configurations and stress triaxialities according to the notch geometry.

| Tested Configurations | N-∞ | N-5 | N-2 |
|---|---|---|---|
| a (refer to Figure 4a) | 6 | 6 | 4 |
| R (refer to Figure 4a) | ∞ | 5 | 2 |
| $\eta_{Brigman} = \frac{1}{3} + \ln(1 + \frac{a}{4R})$ | 0.33 | 0.52 | 0.74 |

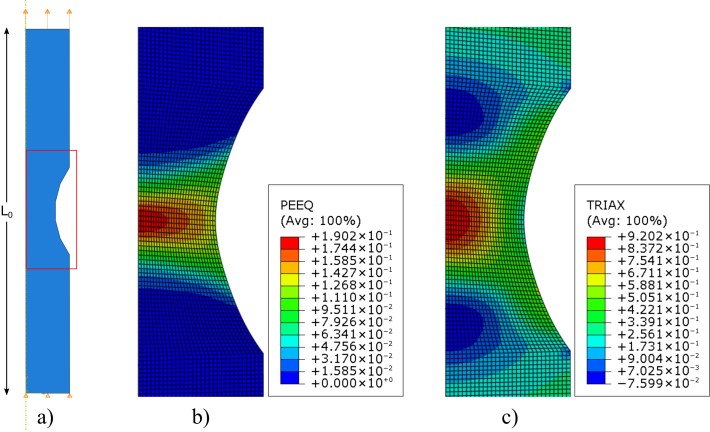

**Figure 5.** (**a**) The boundary conditions on the N-5 specimen geometry; (**b**) the equivalent plastic strain; (**c**) stress triaxiality field distributions for the notch region (signalled red square in the (**a**)).

## 3. Results and Discussion

Figure 6a,b illustrate the micrograph samples (before etching) used for porosity calculation of the AMed and CMed samples, respectively. The relative density values of the AMed maraging steel (99.7%) is in agreement with the manufacturer's specification (99.8%), whereas no porosity was found for the CMed maraging steel. The microstructure of the AMed 18Ni300 chemically etched sample in perpendicular-to-build direction is presented in Figure 6c, and it is in Figure 6d for a CMed sample, where a predominance of plate-like martensitic structure is noticed as well as the absence of thin-lath martensitic structures. The reduced ductility and susceptibility to microcrack formation in plate martensite when compared to lath morphology [38,39] may contribute to the distinct mechanical behaviour between the conventional and AMed maraging steels.

Figure 7a shows an electrolytical etched sample, in perpendicular-to-build direction, where the laser trace is observed. An identically processed metallographic sample in parallel-to-build direction is shown in Figure 7b, where melt pool geometry is evidenced. Grain continuity through melt pools was expected, given that, in Figure 6c, there is no evidence of preferential grain alignment.

Figure 8a presents the tensile tests load–displacement results of the AMed maraging steel. It is clear that there is not only a significant ductility decrease compared to the CMed maraging steel (refer to Figure 8b) but also a lower mechanical strength of the latter metallurgical condition. The strength–ductility ranges are in accordance with the literature [11]. A lower repeatability within the AMed samples is observed. This may be explained by the existence of internal defects in the AMed steels, which can compromise their mechanical behaviour, especially their ductility [40,41]. It is also important to consider the relative size effect of those defects when analysing small material samples, as might have been the case for the tested tensile geometries. In sum, the expected ductility variability of AMed parts is noticeable in the maraging steel printed for the current study.

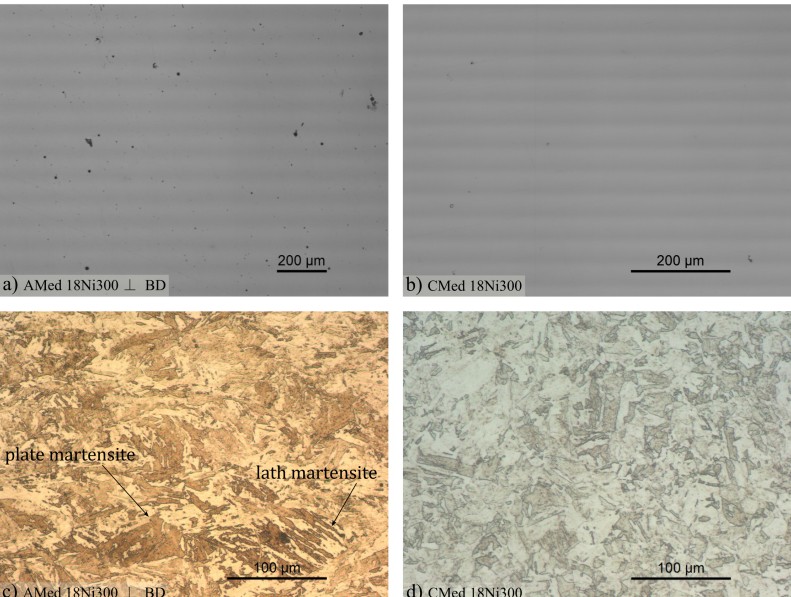

**Figure 6.** Metallographic samples of (**a**) AMed and (**b**) CMed 8Ni300 before etching; chemically etched metallographic samples of (**c**) AMed and (**d**) CMed 18Ni300.

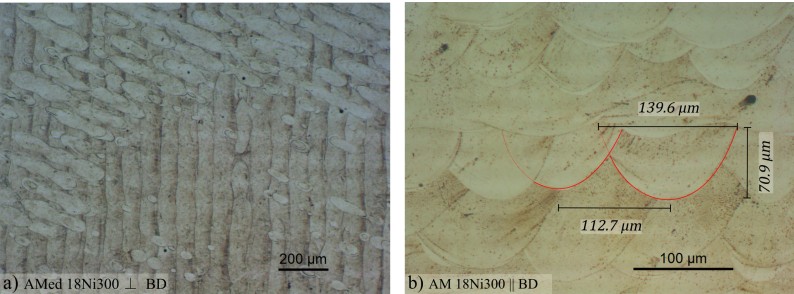

**Figure 7.** Electrolytically etched samples of AMed 18Ni300 in (**a**) perpendicular- and (**b**) parallel-to-build directions (BD).

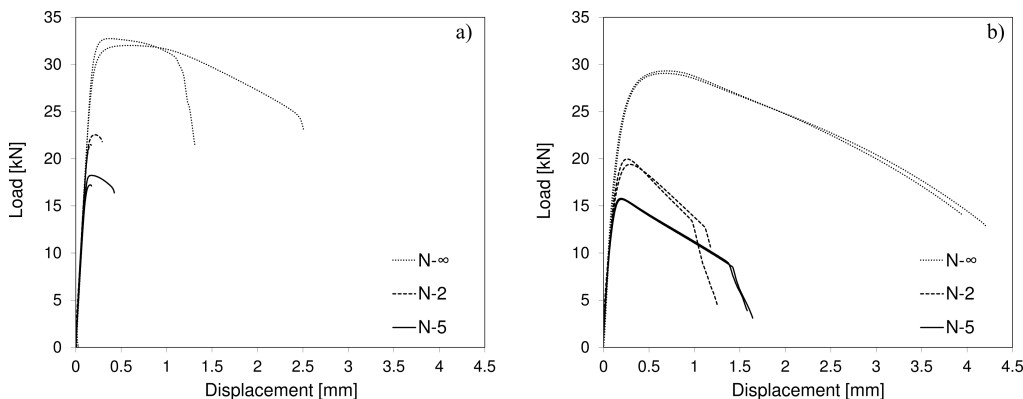

**Figure 8.** Load–displacement curves showing the influence of notch geometry on the tensile behaviour of (**a**) AMed and (**b**) CMed maraging steels.

Given that the tests were conducted for distinct notch geometries, it is also possible to access the stress triaxiality influence on ductility. With regards to notch geometry, the results are in accordance with the predicted stress triaxialities of Table 2. Even though numerical simulation is required to estimate the actual values, a ductility decrease tendency is noticed for stress triaxiality increase.

Necking can be geometrically classified as diffuse or localised [42,43]. While in diffuse necking the flow is quasi-stable due to gradual strain redistribution, in localised necking

flow is unstable, leading to fracture. Diffuse necking seems to occur in maraging steels [44], as can be noticed in the engineering stress–strain curves of the studied materials, presented in Figure 9a. Diffuse necking seems to occur at an early stage and for significant elongation. This is consistent with the very high cross-section area reduction (approximately 80%) in tensile loading, as specified by the manufacturer in the CM metallurgical condition [45]. The specimens' fracture surfaces, as seen in Figure 9b,c, for the CMed material and in Figure 9d,e, for the AMed material, illustrate the necking behaviour on the smooth tensile samples. A much smaller diameter section can be identified for the CMed material, confirming its higher ductility and, thus, higher deformation localisation at fracture (longer diffuse necking regime) than its AMed counterpart. In addition, the typical cup and cone fracture surfaces can be identified for the tested smooth conventional material specimens, confirming its ductile behaviour. These are formed due to the widely known mechanisms of ductile fracture: void nucleation and growth in the central region and shear-dominated separation in the outer region, usually referred to as shear lips [46]. The less pronounced cup and cone fracture geometry in AMed samples seems to show that material fails (abruptly) while flowing in a diffuse necking regime, due to material defects (mostly porosity). In fact, coalesced porosity is highly noticeable, randomly distributed over the AM fracture surfaces (refer to the zoomed in sections of Figure 9d,e).

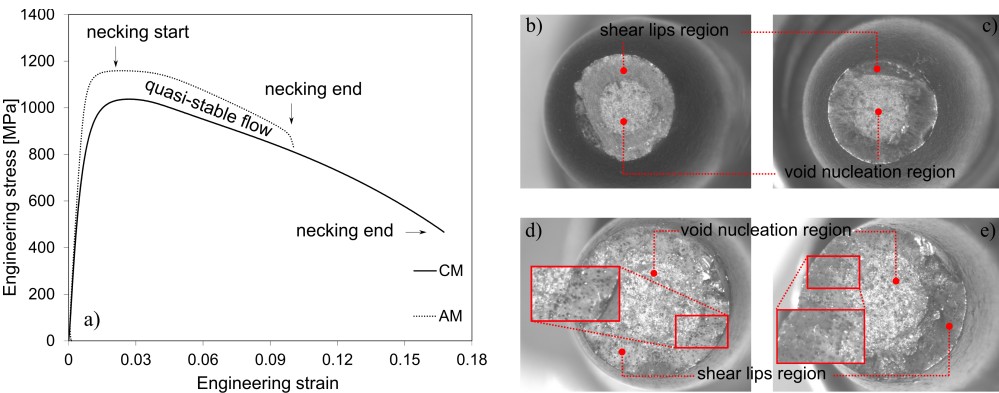

**Figure 9.** (**a**) The engineering stress–strain curve showing quasi-stable flow occurrence (diffuse necking). Fracture surfaces of the tensile test from smooth cylindrical specimens: (**b**) cup and (**c**) cone fracture geometries of CMed maraging steel; (**d**) cup and (**e**) cone fracture geometries of AMed maraging steel.

The material toughness, $U_T$, or their ability to absorb energy up to fracture, was calculated for both AMed and CMed maraging steels. As illustrated in Equation (1), it can be determined through the integration of the tensile stress–strain curve:

$$U_T = \int_0^{\varepsilon_f} \sigma d\varepsilon \tag{1}$$

The obtained results show a $U_T$ in the range of 151–173 mJ/mm$^3$ for the conventional maraging steel, whereas, for the AMed, the higher dispersion leads to a broader range of 95–207 mJ/mm$^3$. Considering that toughness is a direct measure of the strength–ductility ratio, the similar results are coherent with the lower ductility but higher strength of the AMed maraging steel, with regards to its conventional counterpart.

The numerical simulation of the quasi-static tensile tests enables the prediction of pressure dependent fracture strain and, thus, the identification of damage onset behaviour for the high stress triaxiality range ($\geq$0.33). As depicted in Figure 8, the maraging steel shows sensitivity to that parameter, regardless of its metallurgical condition. The analytical stress triaxiality proposed by Bridgman [36] relies on the assumption of rigid perfect plastic flow. Despite being convenient for specimen notch design, that assumption should be corrected by taking into consideration a more realistic constitutive behaviour, resulting in a more accurate representation of the stress triaxiality at the onset of damage. Figure 10

shows the simulated stress triaxiality evolution in function of plastic deformation for both metallurgical conditions of the maraging steel. Despite the accurate analytical prediction for very small strains, significant evolution of stress triaxiality is noticed which is due to incipient geometrical softening (necking) of the specimen. The procedure for fracture strain estimation consisted in matching the load–displacement numerical curves fracture location with experimental load–displacement curves.

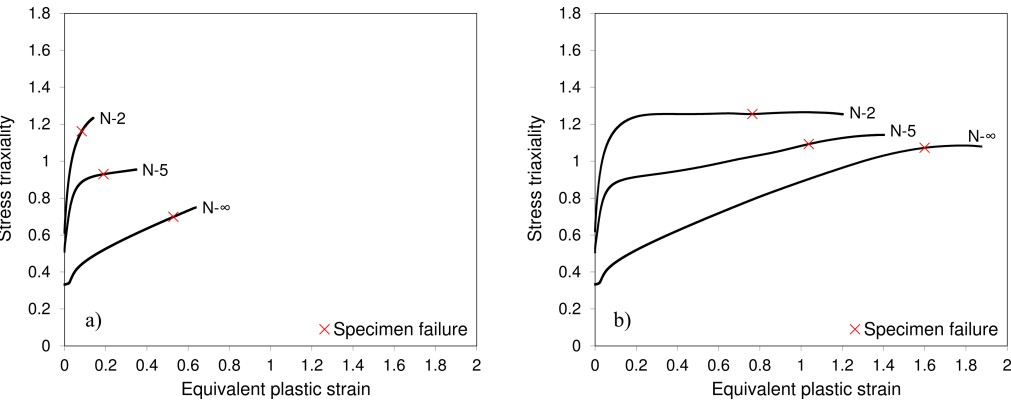

**Figure 10.** Numerically estimated stress triaxiality evolution in function of plastic strain for (**a**) AMed and (**b**) CMed maraging steels.

The load–displacement results of the multiaxial double-notched tests are displayed in Figure 11 for each metallurgical condition. The distinct strength–ductility ratios of AMed and CMed maraging steels are once more in evidence, with the latter showing lower mechanical strength and higher ductility. It is important to note that for the AMed maraging steel all specimens have fractured. However, due to the higher ductility of the CMed maraging steel, only the specimen with a 90° pressure angle has fractured. Figure 12 shows the crack morphology or, in the absence of fracture, the last recorded deformed shape of the specimen for each specimen geometry and metallurgical condition. Higher deformation is noticed in the CMed specimens, based on their shape. Moreover, and as previously pointed out, a marked tendency towards a quasi-stable necking is noticed in the maraging steel, especially in the CMed metallurgical condition. This is highlighted in Figure 12b,d,f, where the excessive material deformation leads to a deviation from the intended state of stress at failure.

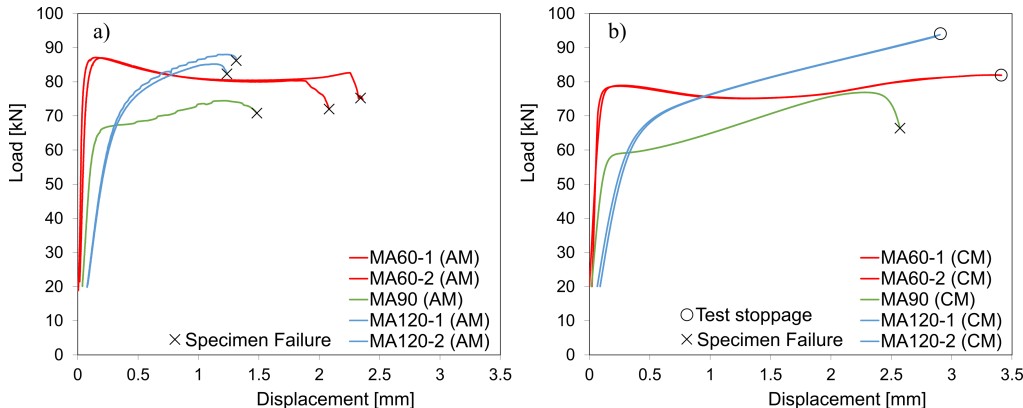

**Figure 11.** Load vs. displacement curves resultant from the multiaxial double-notched tests of (**a**) AMed and (**b**) CMed maraging steels (note: the initial experimental response was removed from the representation due to non-linear contact effects).

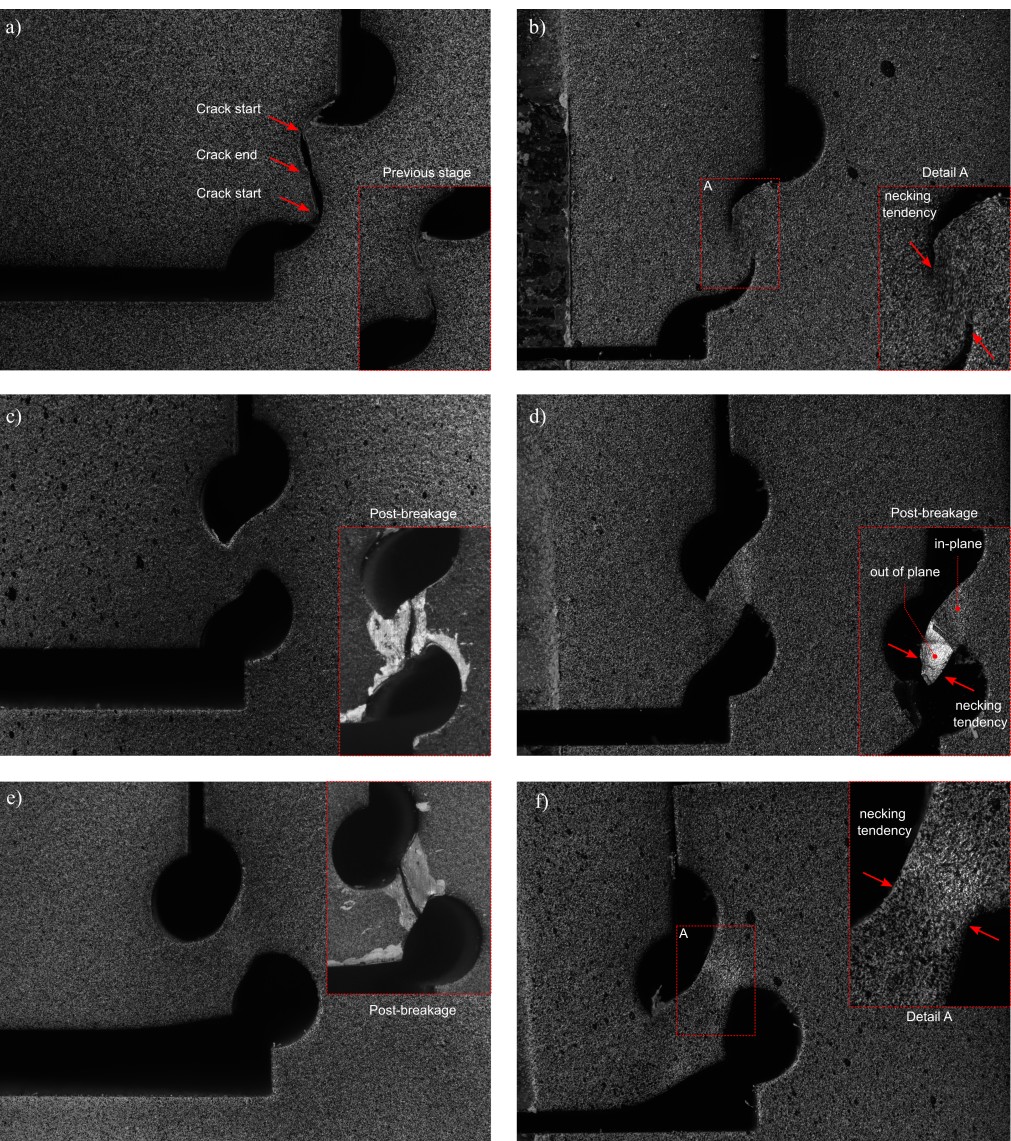

**Figure 12.** Final shape of the multiaxial double-notched 18Ni300 specimens: (**a**) AM 60° showing crack evolution from surface to ligament centre; (**b**) CM 60° showing necking tendency; (**c**) AM 90° showing crack path; (**d**) CM 90° showing necking tendency; (**e**) AM 120° showing crack path; (**f**) CM 120° showing necking tendency.

Figure 12d shows an interesting feature that may jeopardise the validity of the DIC analysis in the conventional material. Fracture seems to occur in different locations depending on in-plane dimension, which may evidence material flow in normal to surface direction. As regards to the AMed specimens, it is observed that, in low stress triaxiality, fracture tends to occur in a more controlled manner, which may be related to the highly negative stress triaxialities that hinder fracture occurrence. This effect is noticed by the visible crack propagation in the specimens with 60° pressure angle (refer to Figure 12a) in contrast with the abrupt failure of specimens with 90° and 120° pressure angles (Figure 12c,e).

The suitability of the multiaxial double-notched specimens towards the definition of plasticity and damage models was pointed out by Abushawashi et al. [24], in the region of low and intermediate stress triaxiality levels, which are not covered by classical tensile notched specimens. Similar to the study in which the specimens were developed, in the current approach, the flow stress was also identified through reverse methodology. Figure 13 presents the used flow stresses for each metallurgical condition. These curves capture the tensile materials' response for low strain values and assume either a negative hardening (AMed) or a perfect plasticity (CMed). Due to allowing for an initial hardening and its

saturation (or even softening), the combined Swift–Voce model (refer to Equation (2)) was used to build the mentioned curves (the resulting constants are presented in Table 3).

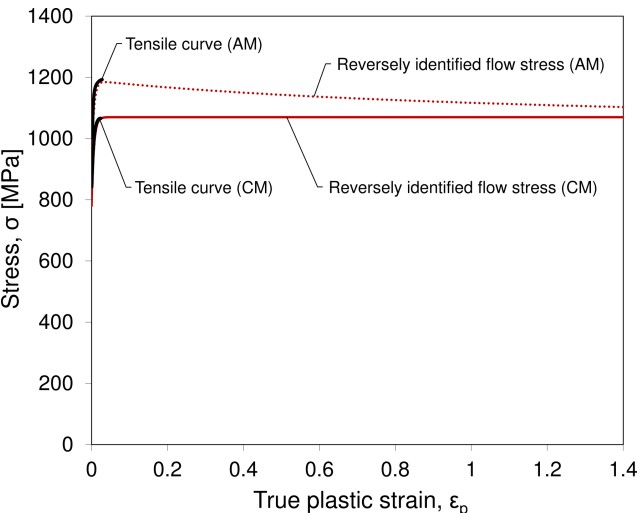

**Figure 13.** Flow stress curves used to simulate the multiaxial specimen for both AMed and CMed metallurgical conditions.

$$\sigma = \alpha \left[ K(\varepsilon_0 + \varepsilon_p)^n \right] + (1 - \alpha) \left[ k_0 - Q(1 - e^{-\beta \varepsilon_p}) \right] \tag{2}$$

**Table 3.** Swift–Voce parameters for both AM and CM inversely identified hardening curves.

| Material | $K$ [MPa] | $\varepsilon_0$ | $n$ | $k_0$ [MPa] | $Q$ [MPa] | $\beta$ | $\alpha$ |
|---|---|---|---|---|---|---|---|
| AM | 950 | 1 | −0.7 | 887.5 | 362.5 | 170 | 0.2 |
| CM | 950 | 1 | 0 | 887.5 | 362.5 | 170 | 0.2 |

The load–displacement results obtained from the simulation of the multiaxial double-notched specimens are shown in Figure 14 for each metallurgical condition. It is important to note that a damage model was not included at this stage, given the initial focus was on the assessment of plastic behaviour. It should also be noted that, for very high displacement values, the softening in numerical prediction of CM MA90 specimen is due to high mesh distortion. When it comes to the comparison of numerical and experimental results, it is observed that the presented flow stress definition results in a maximum deviation of 5% in load prediction and that the typical positive strain-hardening of metals does not seem to represent the maraging steel flow stress. A possible explanation is the occurrence of an early diffuse necking behaviour along with a high ductility of the material may promote a neutral or even negative effective strain-hardening. Shamsdini et al. [47] studied the plastic deformation of AMed maraging steels, stating that strain induced phase transformation under uniaxial tensile loading can also promote strain-hardening variation. The authors claimed that, within the progress of martensitic transformation, there is a tendency for strain-hardening to reach null values, after which load peaks are negative, due to geometric softening, which is coherent with the obtained results.

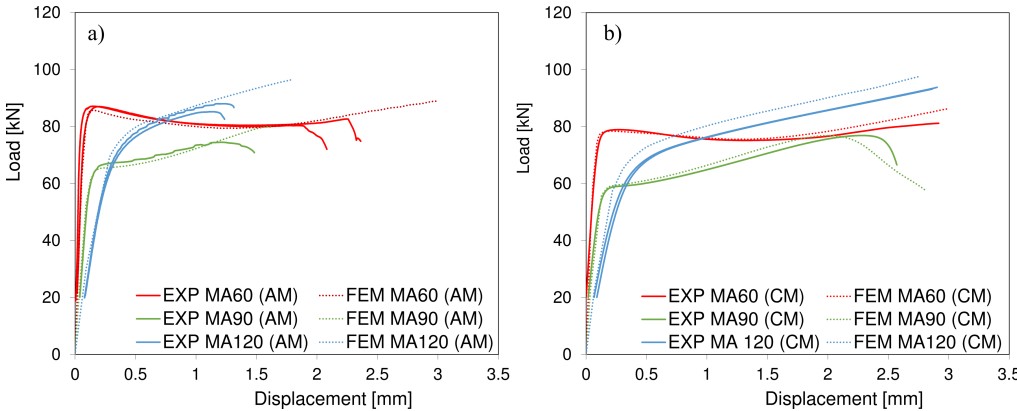

**Figure 14.** Load–displacement results obtained from the simulation of the multiaxial double-notched specimens for both (**a**) AMed and (**b**) CMed metallurgical conditions.

The materials' high strength, ductility and tendency towards geometric softening (out of plane movement) hindered data retrieval from DIC. Figure 15 shows the shear strain fields comparison (between DIC and FEM) of the AMed multiaxial specimen with a pressure angle of 90°. In Figure 15a, which reflects an incipient stage of the test (*d* = 0.2 mm), one can observe a satisfactory correlation not only in the shear strain values but also its localisation in the centre of the specimens' ligament. For a more advanced stage (*d* = 0.5 mm), a highly heterogeneous shear strain field is developed in the numerical model, as seen in the FEM model of Figure 15b. Maximum shear strain seems to localise in two vertical shear bands. Moreover, the specific location where strain is maximum at the contour of the holes seems to realistically portray the crack path, as shown by the red dashed line in Detail A of Figure 15b (refer also to Figure 12c). Despite the inability of DIC to capture the highly heterogeneous strain field, the results show that the DIC predicted values in the centre region of the ligament are somewhat consistent with the average FEM values of the region in between the strain localisation lines (average $\varepsilon_{xy}$ = 0.278), signalled as IBSL in the solid black rectangle line of the detail of Figure 15b. It is also important to highlight that the experimental deformed shapes of the specimens (obtained by DIC) were coherent with the ones obtained by FEM (refer to Figure 16), allowing for further validation of the built numerical models and determined constitutive flow rule.

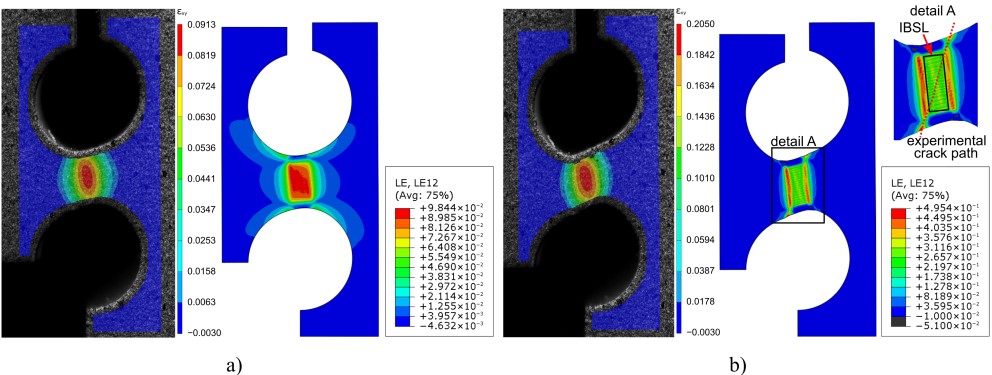

**Figure 15.** Comparison between DIC (left) and FEM (right) shear strain fields at (**a**) an incipient stage (*d* = 0.2 mm) and (**b**) an intermediate stage of the test (*d* = 0.5 mm).

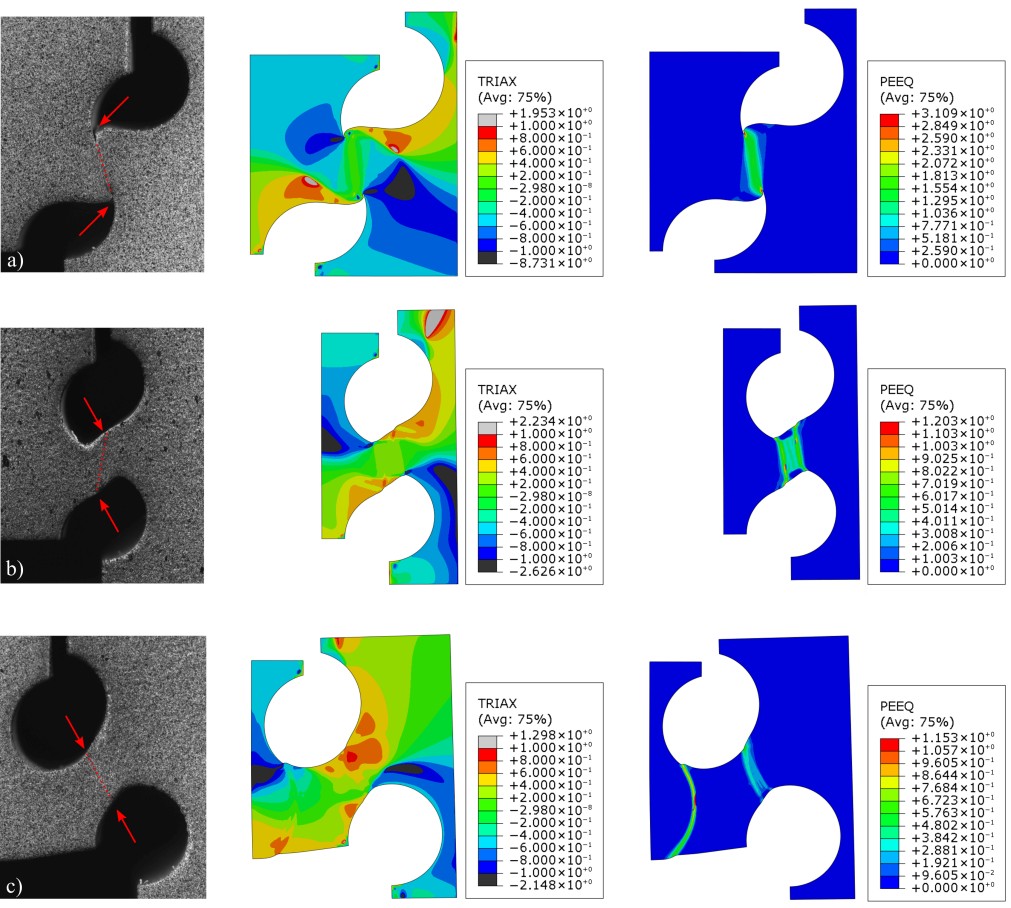

**Figure 16.** Deformed shape comparison of DIC and FEM results and plotted stress triaxiality and equivalent plastic strain of multiaxial double-notched specimens with pressure angles: (**a**) 60°; (**b**) 90°; (**c**) 120°.

Towards the determination of the damage behaviour, the displacement at the onset of damage, $d_f$, was determined for each AMed specimen, through the comparison between experimental and FEM curves (without considering a damage model). It is important to highlight that such procedure was only possible to conduct for the AMed samples, given that the conventional multiaxial double-notched specimens did not fracture, with the exception of one specimen. The fracture displacement was defined as the displacement at which the FEM plastic prediction and the experimental curve diverge. That enabled the determination of the stress triaxiality, $\eta$, and the equivalent plastic strain, $\varepsilon_p^{eq}$, field distributions at damage initiation, which are shown in Figure 16. In the same figure, the crack path is signalled (red dashed straight lines) on the experimental deformed geometries, according to the actual crack location and configuration, as shown in Figure 12. The experimental crack location is in accordance with the maximum $\varepsilon_p^{ep}$ at the hole contour surfaces. This promoted the propagation of the crack from the surface to the centre of the ligament. Moreover, regarding the stress triaxiality, crack seems to occur for maximum positive $\eta$ in the specimens with 90° (Figure 16b) and 120° (Figure 16c) pressure angles. However, due to the prevailing compression/shear loading it occurs for maximum negative $\eta$, in the specimen with a 60° pressure angle, enabling fracture locus determination for distinct stress states. Figure 17a–c shows the $\eta$ and $\varepsilon_p^{eq}$ evolution along the crack path at the onset of damage for specimens with 60°, 90° and 120° pressure angles, respectively. Despite allowing for a relatively easy change on the state-of-stress with a simple modification of notch configuration, the considered multiaxial specimen geometry seems to result in a rather heterogeneous strain field, which makes damage calibration challenging. This is shown above in Detail A of Figure 15b and is now confirmed by the significant evolution of

$\eta$ and $\varepsilon_p^{eq}$ along the crack path. Knowing that crack propagation occurred from the surface to the centre of the ligament, fracture strain, $\varepsilon_f$, and stress triaxiality at fracture, $\eta_f$, were obtained for crack lengths near zero, as plotted in Figure 17d, which, in addition to the $\varepsilon_f$ obtained from tensile specimens, presents the materials' fracture locus. Although it was expected that the multiaxial specimens would allow for fracture locus identification for intermediate pressures, the high ductility along with the geometrical softening of the current material seemed to delay fracture up to a point where the specimen geometry no longer corresponds to the theorised state of stress. However, the results obtained from multiaxial specimens show compatibility with the ones obtained from tensile testing, as higher relative stress triaxialities were expected for the latter. The reduced JC damage law was fitted to the identified exponential decrease in fracture strain, $\varepsilon_f$, for increasing $\eta$. The first term of Johnson–Cook damage law is depicted in Equation (3) and the parameters are shown in Table 4.

$$\varepsilon_f = d_1 + d_2 e^{d_3 \eta} \tag{3}$$

$$G_f = \int_{\varepsilon_0^{pl}}^{\varepsilon_f^{pl}} L\sigma_y d\varepsilon^{pl} \tag{4}$$

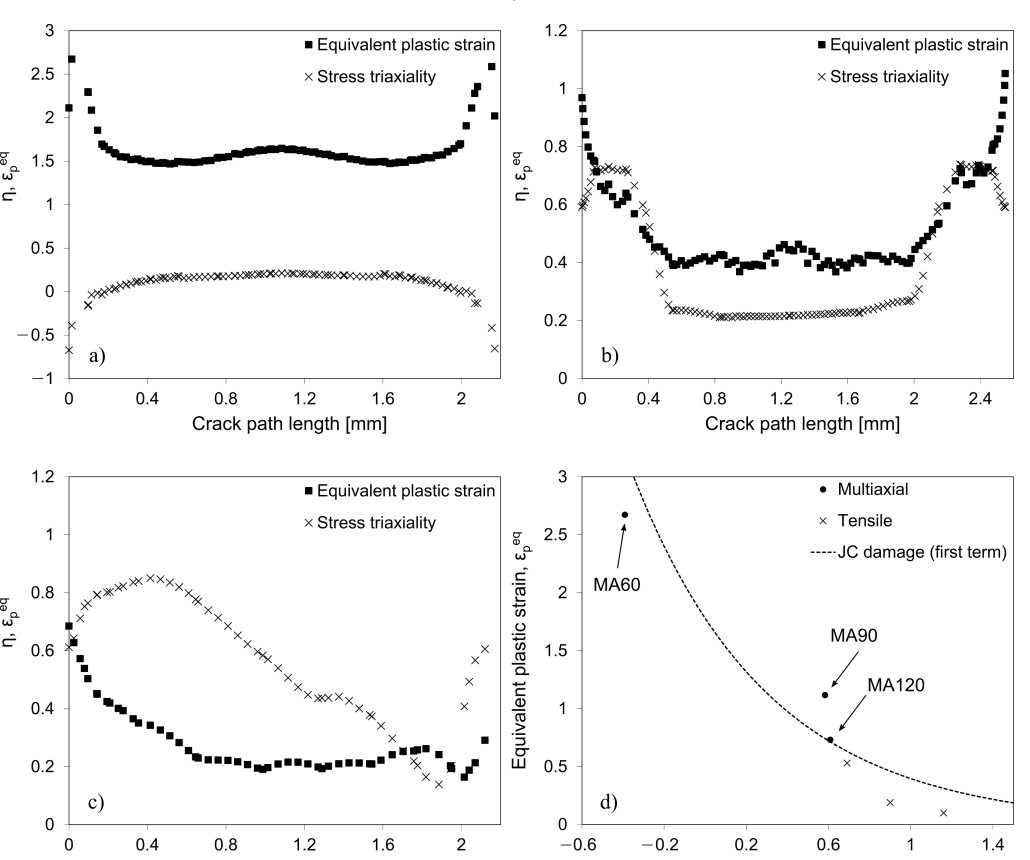

**Figure 17.** Stress triaxiality and equivalent plastic strain evolution along crack path for multiaxial specimens with pressure angles: (**a**) 60°; (**b**) 90°; (**c**) 120°. The identified damage law (**d**) using both multiaxial and tensile specimens.

**Table 4.** First term of Johnson–Cook damage initiation model.

| Material | $d_1$ | $d_2$ | $d_3$ |
|----------|-------|-------|-------|
| AM | −0.01 | 1.77 | −1.5 |

With regards to material damage evolution, element degradation was defined through critical energy dissipation model, as shown in Equation (4). The critical damage dissipation

energy was inversely estimated for the AMed maraging steel, through the comparison between experimental and numerical results. A fracture energy density of $G_f = 10$ mJ/mm$^3$ is proposed. The load–displacement curves of Figure 18 present the comparison between the experimental multiaxial tests and the constitutive modelling (plasticity and damage) for the AMed maraging steel. Taking into account the significantly distinct load levels and fracture strains, a good agreement seems to be found with the suggested approach and results.

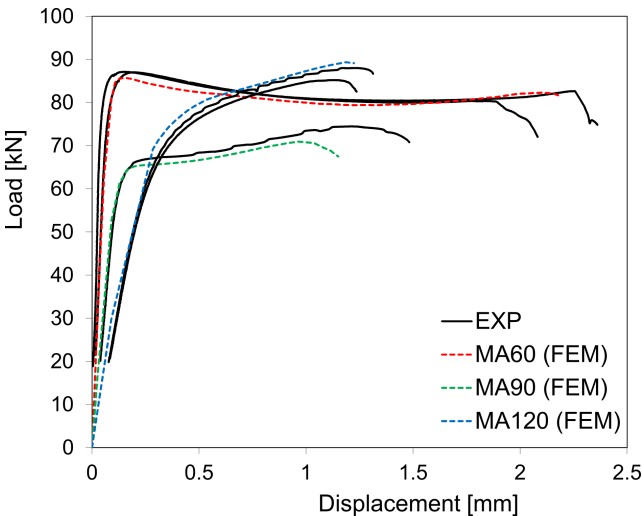

**Figure 18.** Comparison between load vs. displacement curves of experimental and numerical curves (including damage) of the (AMed) double-notched maraging steel within distinct pressure angles.

## 4. Conclusions

The following conclusions can be drawn from this research work:

- Tensile tests showed the higher ductility of the CMed maraging steel compared to the AMed ones. A significant amount of diffuse necking seems to occur in both AMed and CMed maraging steels, which accounts for incipient geometrical softening in tensile conditions.
- The multiaxial double-notched tests confirmed the higher mechanical strength of the AMed metallurgical condition as well as the increased ductility of the conventional maraging steel. In addition, the diffuse necking tendency along with the high ductility of the latter material precluded its fracture characterisation, revealing the inadequacy in selecting the same double-notched geometry towards fracture strain identification, on materials with distinct strength–ductility ratios.
- Even though they are often seen as fracture specimens, the double-notched geometry provided very important insight into material's plasticity and flow stress inverse identification, constituting a valuable alternative to the typical mechanical characterisation methodology that also presents widely known flaws (e.g., friction in compression tests and plastic instability in tensile tests).
- The multiaxial double-notched tests were revealed to be very useful for calibrating the constitutive flow stress behaviour of the AMed maraging steel in plane strain and combined shear–tension/compression conditions. These tests are a useful alternative to typical characterisation approaches (such as compression and tensile tests) which show some recognised limitations (friction and limited strain, respectively). They are also valuable for damage onset definition despite requiring a proper design to avoid large deformations that could lead to unwanted stress states. The capability of the test for evaluation of fracture energies is limited since its does not correspond to a true fracture test, since a significant amount of energy will precede the crack initiation.
- The employed approach using the von Mises isotropic hardening as well as inverse definition of a flow stress with slight softening and uncoupled stress triaxiality sen-

sitive damage initiation was revealed to suitably depict the mechanical response of the AMed 18Ni300 maraging steel under distinct scenarios of stress state (mixed compression and shear, theoretically pure shear and tensile and shear).

- The usage of DIC-levelling approaches, in which the FEA data are processed through the same DIC engine as the experimental DIC data, allows for the mitigation of apparent strain errors, through minimisation of inconsistencies between FEA and DIC, namely the strain calculation algorithm, spatial resolution and data filtering [48]. The fact that such procedure seems to realistically simulate experimental heterogeneous deformations at various load steps [49] may bring a new light on the DIC results. Full-field data can be valuable to allow the search for the proper constitutive model solution in the apparent multiple solution problem.

**Author Contributions:** Conceptualisation, J.X., A.R., P.R. and A.d.J.; methodology, T.S., A.G., J.X., P.R. and A.d.J.; software, T.S., F.S., J.X. and A.d.J.; validation, T.S., F.S., J.X. and A.d.J.; formal analysis, T.S., A.G., F.S., J.X., A.R., P.R. and A.d.J.; investigation, T.S., A.G., F.S., J.X., P.R. and A.d.J.; resources, A.R., P.R. and A.d.J.; data curation, T.S., F.S., J.X., P.R. and A.d.J.; writing—original draft preparation, T.S., J.X. and A.d.J.; writing—review and editing, T.S., A.G., F.S., J.X., A.R., P.R. and A.d.J.; visualisation, T.S., F.S., J.X., P.R. and A.d.J.; supervision, A.R., P.R. and A.d.J.; project administration, A.R., P.R. and A.d.J.; funding acquisition, A.R., P.R. and A.d.J. All authors have read and agreed to the published version of the manuscript.

**Funding:** This work was conducted under the scope of MAMTool (PTDC/EME/31307/2017) and AddStrength (PTDC/EME/31307/2017) projects, funded by Programa Operacional Competitividade e Internacionalização, and Programa Operacional Regional de Lisboa funded by FEDER and National Funds (FCT). This work was supported by FCT, through IDMEC, under LAETA, project UIDB/50022/2020 (UNIDEMI).

**Data Availability Statement:** Not applicable.

**Conflicts of Interest:** The authors declare no conflict of interest.

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
