# Peer review of "Numerical-Experimental Plastic-Damage Characterisation of Additively Manufactured 18Ni300 Maraging Steel by Means of Multiaxial Double-Notched Specimens"

_jmmp, doi:10.3390/jmmp5030084_

Round 1

Reviewer 1 Report

General Comments:

This manuscript presents mechanical characterization of the 18Ni300 maraging steel additively manufactured with laser powder bed fusion. I can tell that the authors know what they’re doing and what they want to say. Therefore, I recommend this manuscript to be accepted for publication. However, my biggest problem was the layout and the writing. It was difficult to get through this manuscript. I understand that English is probably not the native language of some of the authors, but that’s not the main problem. The text contains many unnecessary words and writing with many repetitive and circulatory discussion that detracts from the worth of presented data/discussions. Moreover, organization of data and order of presentation is confusing and makes it difficult for the reader to follow, i.e., flipping back and forth through the manuscript. I have listed some things below that are the “minimum” for the authors to change to help their manuscript be more readable. I hope that with these comments, the authors will be able to edit and refine their results sections and cut down on some of the unnecessary details.

Comments to Address:

  1. English Teacher Comments. I have listed a few that stood out to me the most in specific areas, but I gave up after a while. For more help, MDPI offers “English Editing Services” under the Author Services on their website.
  2. Page 1, Line 2: Include the article “a” before “high degree” – “with a high degree”
  3. Page 1, Line 3: “Inexistent” is not a word, change to “non-existent”
  4. Page 1, Line 3: “recent materials” what is a recent material? Please list some materials/alloys or give a better description of what you mean.
  5. Page 1, Lines 8-9: This sentence is quite a mouthful. I would split it up and use more active language such as “The multiaxial noticed testing demonstrated…”
  6. Page 1, Lines 10-11: This sentence in unnecessary. AM literature is full of these results and subsequently, these sentences. Say something more meaningful about what you did. What stood out the most?
  7. Page 2, Line 41: “In exceptional cases” should be changed to “in some cases”
  8. Page 3, Line 86: “posteriorly” is not a word. Use posterior or “subsequently”
  9. Page 3, Line 103: Not because of the “low melting point” but because of its “high vapor pressure”
  10. Page 3, Lines 113-114: This process is well understood, but after reading this sentence I am not even sure what you did. I would just remove this sentence and say that porosity was measured through defect image analysis, or something similar
  11. Page 3, Line 116: I would go with the classical spelling of “Archimedes”
  12. Page 3, Table 1: Just use a hyphen for reporting a range of compositions instead of using the min/max method.
  13. Page 4, Line 146: change “consist” to “consisted”
  14. Page 7, Line 226: “results is noticed” is not acceptable. Please use terms like observed, found, etc. “It was noticed that” can also be substituted when including observations.
  15. Page 2, Lines 44-63: Get to the point. Lots of unnecessary words and descriptions. I would just delete this entire paragraph unless you can cut it down.
  16. Page 2, Lines 77-83: Delete this introduction to the “Materials and Methods”
  17. Page 3: What machine was this steel printed with in your facility? Please add. Also, change to “Yb fiber laser” in Line 88.
  18. All results on the porosity content and microstructure must be moved to results. We want a concise reporting of what you did in the “Materials and Methods.”
  19. Page 4, Figure 1: I would redo this image. Put side by side images of the CM and AM bother before and after etching. Any additional findings on microstructure that require a higher magnification should be placed into a separate Figure. We need to see the differences between CM and AM first before moving on to something else.
  20. Page 5, Lines 166-170: Cut this paragraph down. We can look up more on DIC if we want to. Just put what is needed.
  21. Page 5, Lines 187-195: Please add a line or two here about Abaqus/CAE. The rest of the paragraph is clear enough.
  22. Page 6, Lines 206-210: Please put in results.
  23. Page 7, Lines 221-232: Please start with microstructure. What did you see? Then discuss the strength and ductility of results instead of basically saying “we did the testing, now let’s move on.”
  24. Page 6, Figure 7: This caption is completely confusing. It is written that this is the “behavior of AM”, then “(a) CM”. What?
  25. The term “CM” was never defined. It does mean “Conventionally Manufactured” I hope? If so, some of the terminology throughout the results conflicts. Again, in Figure 9 caption. What is difference between AM, CM, and maraging steel?

Author Response

The authors would like to thank the reviewers for the valuable comments provided to the original version of the manuscript. Accordingly, all questions have been addressed with the view to improve the manuscript. A revised version of the paper is prepared, in which carefully considerations of all queries mentioned in the reviewers' comments were included. Changes are highlighted in blue colour in the new version of the paper. Please, find enclosed a point-by-point response to the reviewer’s comments.

Comments and Suggestions for Authors
General Comments:
This manuscript presents mechanical characterization of the 18Ni300 maraging steel additively manufactured with laser powder bed fusion. I can tell that the authors know what they’re doing and what they want to say. Therefore, I recommend this manuscript to be accepted for publication. However, my biggest problem was the layout and the writing. It was difficult to get through this manuscript. I understand that English is probably not the native language of some of the authors, but that’s not the main problem. The text contains many unnecessary words and writing with many repetitive and circulatory discussion that detracts from the worth of presented data/discussions. Moreover, organization of data and order of presentation is confusing and makes it difficult for the reader to follow, i.e., flipping back and forth through the manuscript. I have listed some things below that are the “minimum” for the authors to change to help their manuscript be more readable. I hope that with these comments, the authors will be able to edit and refine their results sections and cut down on some of the unnecessary details.
R: Dear reviewer, thanks for the comments and the opportunity given to the authors to improve the manuscript. The manuscript was comprehensively revised in order to improve the English. Further the structure of the paper was rearranged in order to make it better organized. In particular, a clear separation between the methods and results was performed.
Comments to Address:
English Teacher Comments. I have listed a few that stood out to me the most in specific areas, but I gave up after a while. For more help, MDPI offers “English Editing Services” under the Author Services on their website.
Page 1, Line 2: Include the article “a” before “high degree” – “with a high degree”
Authors: This sentence has been modified in the revised manuscript accordingly.
Page 1, Line 3: “Inexistent” is not a word, change to “non-existent”
Authors: This sentence has been modified in the revised manuscript accordingly.
Page 1, Line 3: “recent materials” what is a recent material? Please list some materials/alloys or give a better description of what you mean.
Authors: A more specific description of the materials has been included. The goal was to refer to the additively manufactured materials.
Page 1, Lines 8-9: This sentence is quite a mouthful. I would split it up and use more active language such as “The multiaxial noticed testing demonstrated…”
Authors: The sentence was shortened and more active language was used.
Page 1, Lines 10-11: This sentence in unnecessary. AM literature is full of these results and subsequently, these sentences. Say something more meaningful about what you did. What stood out the most?
Authors: The sentence has been modified into something more meaningful concerning the conducted work.
Page 2, Line 41: “In exceptional cases” should be changed to “in some cases”
Authors: This sentence has been modified in the revised manuscript accordingly.
Page 3, Line 86: “posteriorly” is not a word. Use posterior or “subsequently”
Authors: This sentence has been modified in the revised manuscript accordingly.
Page 3, Line 103: Not because of the “low melting point” but because of its “high vapor pressure”
Authors: This sentence has been modified in the revised manuscript accordingly.
Page 3, Lines 113-114: This process is well understood, but after reading this sentence I am not even sure what you did. I would just remove this sentence and say that porosity was measured through defect image analysis, or something similar
Authors: This sentence has been replaced by a simpler description, as suggested.
Page 3, Line 116: I would go with the classical spelling of “Archimedes”
Authors: This sentence has been modified in the revised manuscript accordingly.
Page 3, Table 1: Just use a hyphen for reporting a range of compositions instead of using the min/max method.
Authors: The table has been modified in the revised manuscript accordingly.
Page 4, Line 146: change “consist” to “consisted”
Authors: This sentence has been modified in the revised manuscript accordingly.
Page 7, Line 226: “results is noticed” is not acceptable. Please use terms like observed, found, etc. “It was noticed that” can also be substituted when including observations.
Authors: This sentence has been modified in the revised manuscript accordingly.
Page 2, Lines 44-63: Get to the point. Lots of unnecessary words and descriptions. I would just delete this entire paragraph unless you can cut it down.
Authors:  This paragraph focuses on the necessity in characterizing additively manufactured materials in other conditions than the most common uniaxial testing. In addition, a brief review on the possible configurations and setups of notched specimens tests is performed. Moreover, the authors were asked to include more information on notch work by other reviewer, which highlights the importance of this particular theme. In sum, more references were added referring to contemporary studies on notched-specimens testing.
Page 2, Lines 77-83: Delete this introduction to the “Materials and Methods”
Authors: The introduction has been deleted.
Page 3: What machine was this steel printed with in your facility? Please add. Also, change to “Yb fiber laser” in Line 88.
Authors: All information regarding the processing condition of the additively manufactured samples have been included in the manuscript. With regards to the equipment’s’ reference, the authors would prefer not to disclose such information, in order to avoid any linkage/bond of these samples results with commercial AM manufacturers.
The sentence of line 88 has been modified in the revised manuscript accordingly.
All results on the porosity content and microstructure must be moved to results. We want a concise reporting of what you did in the “Materials and Methods.”
Authors: All results on the porosity content and microstructure have been moved to results section, as suggested.
Page 4, Figure 1: I would redo this image. Put side by side images of the CM and AM bother before and after etching. Any additional findings on microstructure that require a higher magnification should be placed into a separate Figure. We need to see the differences between CM and AM first before moving on to something else.
Authors: This figure has been modified in the revised manuscript accordingly. Metallographic images of the samples before and after chemical etching are shown side by side. Another figure was added showing the electrolytic etched AM samples, in two orientations (parallel and perpendicular) to build direction, showing the typical AM macrostructures (melt pool and laser trace).
Page 5, Lines 166-170: Cut this paragraph down. We can look up more on DIC if we want to. Just put what is needed.
Authors: This paragraph has been modified in the revised manuscript accordingly.
Page 5, Lines 187-195: Please add a line or two here about Abaqus/CAE. The rest of the paragraph is clear enough.
Authors: This sentence has been modified in the revised manuscript accordingly.
Page 6, Lines 206-210: Please put in results.
Authors: The paragraph of lines 206-210 concerns the adopted methodology as well as the major advantages in using it for the purpose of the current study. Accordingly authors do not consider the placement of such details in the results section.
Page 7, Lines 221-232: Please start with microstructure. What did you see? Then discuss the strength and ductility of results instead of basically saying “we did the testing, now let’s move on.”
Authors: The results and discussion section has been revised in order to start with the description and analysis of the microstructure. Authors have compared two material conditions and looked into the differences and discussed it according to the existing literature (44 citations). Metallurgical considerations supported the mechanical performance of the materials.
Page 6, Figure 7: This caption is completely confusing. It is written that this is the “behavior of AM”, then “(a) CM”. What?
Authors: This caption has been modified in the revised manuscript accordingly.
The term “CM” was never defined. It does mean “Conventionally Manufactured” I hope? If so, some of the terminology throughout the results conflicts. Again, in Figure 9 caption. What is difference between AM, CM, and maraging steel?
Authors: The nomenclature has been revised in the whole document (and the nomenclature table was edited). The caption of fig.9 has been modified in the revised manuscript.

Reviewer 2 Report

The authors present an interesting work on notch study. A few comments are given below

  1. The literature review should include details about notch work. Recently published works in notches are not limited to AM materials. Look at Albinmousa et al, Jahed et al, Vervami et al etc
  2.  Elastoplastic ABAQUS approach is not the best approach to model notch samples. Why did the authors use this method ?
  3.  Which LPBF machine was used with 70 um spot size ?
  4.  Where was the material bought for LPBF ?
  5.  How are samples prepared before testing ? they look extremely smooth ?
  6.  Why are all samples showing softening ?(Fig 7) This is not the expected behavior
  7.  Please explain where did this occur ? -> "coalesced porosity is highly noticeable"
  8.  Fig 10 why is it not starting from ( 0,0)?
  9.  Line 407 - how do these results provide insight into the material - The arguments are not clear
  10.  Conclusions are not written well and are not inline with the results in the paper. Numerical results should be discussed in the conclusions along with resultsand observations.

Author Response

The authors would like to thank the reviewers for the valuable comments provided to the original version of the manuscript. Accordingly, all questions have been addressed with the view to improve the manuscript. A revised version of the paper is prepared, in which carefully considerations of all queries mentioned in the reviewers' comments were included. Changes are highlighted in blue colour in the new version of the paper. Please, find enclosed a point-by-point response to the reviewer’s comments.

Comments and Suggestions for Authors
The authors present an interesting work on notch study. A few comments are given below
The literature review should include details about notch work. Recently published works in notches are not limited to AM materials. Look at Albinmousa et al, Jahed et al, Vervami et al etc
Authors: The literature review section in which some details about notch work is focused has been enriched with some works and corresponding references by the suggested authors.
Elastoplastic ABAQUS approach is not the best approach to model notch samples. Why did the authors use this method?
Authors: The main goal of the current study regards the identification of plastic and damage laws that closely portray the mechanical response of the additively manufactured 18Ni300 maraging steel, in distinct stress-state conditions. For that, uncracked notched specimens were employed towards inverse identification of damage initiation, its evolution as well as the flow stress. It should be highlighted that fracture mechanics is not the focus of the work (i.e. crack propagation behaviour). Instead, the problem can be regarded from the continuum mechanics perspective, considering that no singularities at the root notch are expected. Therefore, an elastoplastic analysis with an uncoupled damage initiation and evolution accounting for inelastic behaviour on notch root seems, to the author’s knowledge, a perfectly valid and appropriate approach.
Which LPBF machine was used with 70 um spot size ?
Authors:  The authors would prefer not to disclose such information, in order to avoid any linkage/bond of these samples results with the AM manufacturer.
Where was the material bought for LPBF ?
Authors:  The powder material is of the same manufacturer of the LPBF equipment. The samples were processed by the manufacturer according to best printing conditions, regarding mechanical strength. Size statistics of the powder was provided in the manuscript.
How are samples prepared before testing ? they look extremely smooth ?
Authors: Samples were machined before testing in order to ensure the correct dimensional and geometrical tolerances of the specimens. That information has been included in the manuscript.
 Why are all samples showing softening ?(Fig 7) This is not the expected behavior
Authors: Figure 7 shows the load-displacement results, for distinct notched and smooth configurations of both additively and conventionally manufactured 18Ni300 maraging steel. Since the figure is showing load and displacement (and not true-strain vs. true stress) the “softening” is apparent and is related with the necking tendency of the maraging steel, which seems to occur in a quasi-stable manner. Anyway, the inversely identified flow stress for the AM maraging steel shows some softening degree, which is in accordance with the high area reduction of this particular material, as indicated in references [37], [38] and [12].
 Please explain where did this occur ? -> "coalesced porosity is highly noticeable"
Authors: Coalesced porosity is visible in the fracture surfaces of the additively manufactured tensile samples. The manuscript has been modified in the revised manuscript in order to more accurately indicate where this phenomenon occurs. Zoom-in windows in figures 9d and 9e have been included to better show the coalesced porosity occurrence.
 Fig 10 why is it not starting from ( 0,0)?
Authors: The deviations to parallelism (contact between the machine plates and the specimens) in the beginning of the experimental tests motivates a non-linear response that has no significantly physical meaning. Curves were shown from linear part (with elastic response aligned with origin).
 Line 407 - how do these results provide insight into the material - The arguments are not clear
Authors: The double-notched specimens provided very relevant data as they allowed for the inverse identification of flow stress, damage initiation and evolution of the additively manufactured maraging steel. That particular conclusion refers to the valuable and more thorough alternative that these tests constitute, when compared with more conventional testing, such as tensile and/or compression. Therefore, the authors are claiming that this research allows better understanding (insight) of material mechanical behaviour.

Conclusions are not written well and are not inline with the results in the paper. Numerical results should be discussed in the conclusions along with results and observations.
Authors: Conclusions were revised since the numerical aspects were not properly considered in the original manuscript. In the revised manuscript a specific conclusion was added regarding the numerical results.

Reviewer 3 Report

The study of the mechanical properties of materials obtained using modern technologies of additive manufacturing is an important scientific and practical task. To solve these problems, the authors use traditional and modern methods of experimental research in combination with numerical modeling. In this regard, I believe that the work is relevant.

Having studied the manuscript in detail, I believe that the following points should be highlighted:

  1. Introduction.

The literature review consists mainly of contemporary sources.

Comment. The purpose of the study is not specified. At the end of the section, the authors state what they have done. Instead, the specific purpose or goals of the work should be stated.

  1. Materials and methods.

The research technique was chosen adequately.

Comment. Line 88 indicates that the optimized print settings were used. At the same time, it is not indicated exactly how the optimization was carried out. The criteria for density and strength are clear, but it is not clear from this work how well the optimization was performed (how many samples were made, which set of modes was analyzed, etc.). Maybe the authors have done optimization research before? If so, then it is worth providing a link to this work.

  1. Results and discussion.

The results are described in great detail and beautifully presented.

The discussion is quite detailed, supported by references to literary sources.

  1. Conclusions.

The presented conclusions are consistent with the results presented in Section 3.

Author Response

The authors would like to thank the reviewers for the valuable comments provided to the original version of the manuscript. Accordingly, all questions have been addressed with the view to improve the manuscript. A revised version of the paper is prepared, in which carefully considerations of all queries mentioned in the reviewers' comments were included. Changes are highlighted in blue colour in the new version of the paper. Please, find enclosed a point-by-point response to the reviewer’s comments.

Comments and Suggestions for Authors
The study of the mechanical properties of materials obtained using modern technologies of additive manufacturing is an important scientific and practical task. To solve these problems, the authors use traditional and modern methods of experimental research in combination with numerical modeling. In this regard, I believe that the work is relevant.

Having studied the manuscript in detail, I believe that the following points should be highlighted:
Introduction.
The literature review consists mainly of contemporary sources.
Comment. The purpose of the study is not specified. At the end of the section, the authors state what they have done. Instead, the specific purpose or goals of the work should be stated.
Authors: A more clear definition of the purpose of the study has been included at the end of the introduction, stating the specific goals of the current study.

Materials and methods.
The research technique was chosen adequately.
Comment. Line 88 indicates that the optimized print settings were used. At the same time, it is not indicated exactly how the optimization was carried out. The criteria for density and strength are clear, but it is not clear from this work how well the optimization was performed (how many samples were made, which set of modes was analyzed, etc.). Maybe the authors have done optimization research before? If so, then it is worth providing a link to this work.
Authors: The optimization of the printing parameters was performed by the LPBF manufacturer for the used LPBF machine and metal powder. In other words, authors have used the best practice know for this material and respective processing parameters have been provided.
Results and discussion.
The results are described in great detail and beautifully presented.
The discussion is quite detailed, supported by references to literary sources.
Conclusions.
The presented conclusions are consistent with the results presented in Section 3.
Authors: Thanks for positive comments and mostly for understanding our work. Anyway he tried to improve even more the paper taking into account all reviewer comments.

Round 2

Reviewer 1 Report

The authors have addressed the comments made by this reviewer, which allows for acceptance of the manuscript in its current form.